# Adaptive Fuzzy Fault-Tolerant Formation Control For Third-Order Heterogeneous Vehicle Platoon System With Intermittent Actuator Faults

1st Zhiting Zhou
*College of Science*
*Liaoning University of Technology*
Jinzhou, China
15042276231@163.com

2nd Kewen Li
*College of Science*
*Liaoning University of Technology*
Jinzhou, China
likewen2018@163.com

3rd Yongming Li*
*College of Science*
*Liaoning University of Technology*
Jinzhou, China
l_y_m_2004@163.com

*Abstract*—This paper investigates the problem of prescriptive performance tolerant control of a heterogeneous vehicle platoon with intermittent actuator failures. A finite time prescribed performance function is then suggested, which makes use of the prescribed performance control (PPC) technique. This can prevent collisions and maintain connectivity between nearby vehicles while guaranteeing that the platoon system's tracking error converges to a predefined region within a prescribed amount of time. Then, in the context of a sliding mode control approach, an adaptive fault-tolerant platoon control scheme is proposed by incorporating a fuzzy logic system (FLS) to compensate for the effect of actuator faults. This system guarantees that all signals in the closed-loop system are actually finite time stable and demonstrates the ability to ensure both the individual vehicle stability and string stability. And finally, a four-vehicle platoon is numerical simulation was carried out to show the effectiveness of the proposed algorithm.

*Index Terms*—Heterogeneous vehicle platoon control, Finite-time control, Prescribed performance control, Intermittent actuator faults.

## I. Introduction

Intelligent transportation systems are now a reality thanks to the recent rapid advancements in automatic control and communication technologies. One of the most useful methods for reducing traffic congestion, energy consumption, and improving transportation efficiency is vehicle platoon control, which also has major benefits for improving traffic conditions [1], including lowering tailpipe emissions, boosting traffic throughput, and improving highway safety.

One of the essential elements of intelligent transportation systems that is receiving more and more attention is vehicle platoon management. In addition to lowering air resistance and fuel consumption, following the lead car in the same lane at a close distance also increases road traffic efficiency and safety. Maintaining a specific inter-vehicle spacing is crucial for string stability, which is the stability of the convoy as a whole [2]. This may be achieved by efficiently controlling each vehicle in the convoy. The stability of traffic flow, string stability, and traffic efficiency are all directly impacted by the shop spacing strategy that is chosen.

Since it is essential to the effectiveness and stability of fleet control, the topology of information flow between vehicles is another crucial component of fleet management. Different information flow topologies have an impact on the way the fleet maneuvers and communicates while driving. Appropriate feedback controllers are needed when designing for various topologies in order to guarantee system efficiency and stability.

The unpredictability of actuator faults in terms of time, mode, and amplitude makes it an important issue in the field of control engineering to mitigate the effects of these faults on the control system [4]. Adaptive control is a useful method of compensating for actuator faults because it automatically adjusts to the uncertainty introduced by the faults, using the adaptive law to adjust the controller parameters online to achieve the desired control objective even in the absence of known fault information [5]. In practice, actuators in control systems usually suffer from unstable intermittent faults, often alternating between different fault states and normal functions [6]. How to efficiently preserve constrained parameter estimates and closed-loop system stability is one of the primary technological challenges in the field of fault compensation for occasionally malfunctioning actuators. Actuator fault compensation for intermittent failures in nonlinear systems is currently the subject of few investigations. The majority of research has concentrated on guaranteeing boundedness in the sense of mean-square deviation rather than addressing the correction of intermittent actuator defects in certain kinds of nonlinear systems, and no clear link has been found between tracking error and design parameters. How to completely avoid the impact of actuator failures on a control system has become a technical challenge in the field of control due to the fact that the moment of failure, the mode of failure and the value of the failure are all completely unexpected.

The issue of intermittent actuator fault compensation for nonlinear systems was further investigated by Lai et al. [7]. An uncertain time lag in nonlinear systems including intermittent actuator fault compensation was the subject of literature [8].

This work is supported by National Natural Science Foundation of China under Grant U22A2043

In spacecraft attitude control systems, the literature [9] concentrated on compensating for intermittent actuator defects. K-filtering approaches have been utilized within an adaptive output feedback control system [10], primarily to offset the consequences of intermittent actuator failures [11]. The aforementioned research, however, have only shown the system's stability and steady-state tracking metrics; they haven't established a connection between the system's transient metrics and the design parameters in terms of tracking error. Therefore, changing the design parameters will not result in an improvement in the system's transient performance.

In the platoon control of vehicles. It is implausible that the majority of research findings currently available can only attain asymptotically stable tracking performance. Finite time control guarantees a restricted convergence time while also offering faster convergence and more robust disturbance suppression. This study applies finite time theory to fleet control [12]. As a result, the spacing error cannot converge to zero in a finite amount of time. Rather, it can only ensure reaching the sliding surface in a given amount of time. A different approach that has been suggested uses distributed control to accomplish tracking in a finite amount of time, however it disregards the system's string stability. Therefore, creating a coordinated vehicle control strategy that works well and permits the spacing error to converge to zero in a finite amount of time is a difficult research problem.

This study attempts to address the prescribed performance control (PPC) problem for heterogeneous vehicle platooning systems that concurrently account for intermittent actuator faults. Using terminal sliding mode control (TSMC), FLS and PPC. Below is a summary of the work's principal contributions:

1) For vehicle platoon with actuator failures, we present an FTC strategy in this study. The impacts of intermittent actuator failures are mitigated by the suggested FTC approach. The suggested FTC approach is unable to address the issue of intermittent actuator problems, despite the authors' additional research on actuator faults in the literature [1].

2) As opposed to [2], the suggested strategy in this work, which makes use of the PPC technique, always restricts the tracking error to a predetermined set of arbitrarily tiny residuals, independent of the actuator's periodic failures. This means that the required system's transient and steady state performance may be determined in terms of tracking error.

The paper is structured as follows. Section 2 presents system modeling and problem formulation. Section 3 gives the main results. In Section 4, numerical simulations are given to illustrate the performance of the proposed control method, which followed by the conclusion in Section 5.

## II. SYSTEM MODELING AND PROBLEM FORMULATION

### A. Vehicle Longitudinal Dynamics

Consider a platoon of vehicles that are controlled longitudinally, consisting of a leader and N followers. The leader's kinematics are represented by a certain model.

$$\dot{p}_0(t) = v_0(t), \dot{v}_0(t) = a_0(t) \tag{1}$$

where $p_0(t)$, $v_0(t)$ and $a_0(t)$ represent the position, velocity and acceleration information of the lead vehicle. The motion and dynamics model of each follower $i$ (where $i \in 1, 2, \cdots, N$) can be represented by a third-order nonlinear equation with uncertainties.

$$\begin{cases} \dot{p}_i(t) = v_i(t), \\ \dot{v}_i(t) = a_i(t), \\ \dot{a}_i(t) = \frac{1}{m_i \tau_i} u_{ai} - f_i(v_i, a_i) + d_i(t), \end{cases} \tag{2}$$

where

$$f_i(v_i, a_i) = \frac{1}{m_i \tau_i} [\rho_a A_i C_{ai}(\frac{1}{2} v_i^2 + \tau_i v_i a_i) + d_{mi}] + \frac{1}{\tau_i} a_i \tag{3}$$

where $p_i(t)$, $v_i(t)$ and $a_i(t)$ respectively are the real-time position, velocity and acceleration of the $i$ th following vehicle, $u_{ai}$ denotes the control input of the $i$-th car, $d_i(t)$ denotes the unknown external disturbance caused by wind, road condition, etc. $m_i$ denotes the mass of the $i$-th vehicle, $\tau_i$ denotes the time constant of the vehicle's engine, $d_{mi}$ denotes mechanical resistance, $\rho_a$ denotes the air mass, $C_{ai}$ denotes the vehicle's cross-sectional area, and $A_i$ denotes the coefficient of aerodynamic drag. Due to the technological constraints, the parameters, such as $\tau_i$ not be obtained accurately and thus $f_i(v_i, a_i)$ is unknown. It should be noted that a platoon is classified as homogeneous if all vehicles have the same dynamics, while it is considered heterogeneous if the vehicles have different dynamics, such as $m_i$, $\tau_i$ are different in the same case.

### B. CTH Policy

To enhance platoon security and stability, this paper employs the constant time headway policy (CTHP) and defines the tracking error of adjacent vehicle spacing as

$$\begin{cases} d(t) = p_{i-1}(t) - p_i(t) - L_i \\ d^*(t) = \Delta_i + h_i v_i(t) \end{cases} \tag{4}$$

$$e_i(t) = p_{i-1}(t) - p_i(t) - L_i - \Delta_i - h_i v_i(t) \tag{5}$$

The length of each vehicle i is denoted by $L_i$ for i=0,1,...,n, and the minimum safe distance between vehicles is denoted by $\Delta_i, h_i$ is the distance between vehicles when traveling. The spacing between workshops, denoted as $d^* \in \mathbb{R}_+$, is the desired distance when a vehicle comes to a stop.

The first-order derivative of $e_i(t)$ is

$$\dot{e}_i(t) = v_{i-1}(t) - v_i(t) - h_i a_i(t) \tag{6}$$

## C. Fuzzy Logic System

For a continuous function $f(x)$ defined on a bounded and compact set $\Omega$, there exists a condition ensuring $f(x)$ stays within $\varepsilon > 0$ of its values.

$$\sup_{x \in \Omega} |f(x) - \theta^T \varphi(x)| \leq \varepsilon \quad (7)$$

Where $x = [x_1, \ldots, x_j]^T$ denote the input of the Fuzzy Logic System (FLS), with $\varepsilon$ representing the fuzzy minimum approximation error. The weight vector $\theta^T$ is composed of elements $\theta_1, \theta_2, \ldots, \theta_n$. Here, $\varphi(x) = [\varphi_1(x), \ldots, \varphi_n(x)]^T$ serves as the vector of fuzzy basis functions, and $\varphi_i(x_i)$ is chosen as

$$\varphi_i(x_i) = \frac{\prod_{i=1}^{j} \mu_{F_i^l}(x_i)}{\sum_{l=1}^{n} (\prod_{i=1}^{j} \mu_{F_i^l}(x_i))} \quad (8)$$

## D. Intermittent Actuator Fualt Model

The inetrmitten acuator faults are considered in system model, it can be expressed as

$$\begin{cases} \mathbf{u}_{ai}(t) = \Xi_{ki} u_i(t) + r_{ki} \\ \Xi_{ki} r_{ki} = 0 \end{cases} \quad t \in [t_{ki,h}, t_{ki,e}] \quad (9)$$

where $\Xi_{ki} \in [0,1]$, with $k$ denoting the $k$th fault model. Meanwhile, $r_{ki}$ represents the unknown stuck fault bias, adhering to the constraint $|r_{ki}| \leq \bar{r}_{ki}$ where $\bar{r}_{ki}$ is a constant with $\bar{r}_{ki} > 0$. Additionally, $t_{ki,h}$ and $t_{ki,e}$ signify the start and end times of the $k$th controller failure. By understanding the configuration of these parameters, we gain insight into the behavior and failure scenarios of the controller under different fault models.

Three distinct scenarios can be categorized under model (9).

1) When $\Xi_{ki} = 1$ and $r_{ki} = 0$, it indicates that the actuator is in normal case.
2) If $0 < \underline{\Xi}_{ki} \leq \Xi_{ki} \leq \bar{\Xi}_{ki} < 1$ and $r_{ki} = 0$, then it implies the actuator with the partial loss of effectiveness. For example, $\Xi_{ki}=0.6$ represents the actuator loses its 40% effectiveness.
3) $\Xi_{ki} = 0$ and $r_{ki} \neq 0$, which means that the actuator faults occur, at least one actuator is ensured to be faultless.

Further, (2) can be written as

$$\begin{cases} \dot{p}_i(t) = v_i(t), \\ \dot{v}_i(t) = a_i(t), \\ \dot{a}_i(t) = \frac{1}{m_i \tau_i} (\Xi_{ki} u_i(t) + r_{ki}) - f_i(v_i, a_i) + d_i(t), \end{cases} \quad (10)$$

**Assumption 1** ([1]). Disturbances $d_i(t)$ for $i = 1, \ldots, N$ are bounded, with $|d_i(t)| \leq \mathbf{d}_i^*$, where $\mathbf{d}_i^*$ is the unknown maximum value each can reach.

## E. Control Objectives

The following illustrates the goal of the heterogeneous vehicular platoon.

(1) Individual vehicle stability ([6]): The tracking error between neighboring vehicles converges in finite time converging to a small neighborhood near the origin:

$$\lim_{t \to T_r} |e_i(t)| \leq \varepsilon_i \quad (11)$$

where $T_r$ is the tracking error convergence time, and $\varepsilon_i$ is the smaller positive number.

(2) String stability ([6]): For vehicle platoon, the string stability can be described mathematically as follows:

$$|e_i(x)| \leq |e_{i-1}(x)| \leq \ldots \leq |e_1(x)| \quad (12)$$

**Lemma 1** ([12]). For variables $\varepsilon_1, \varepsilon_2, \ldots, \varepsilon_n > 0$, there is

$$\sum_{i=1}^{n} \varepsilon_i^p \geq (\sum_{i=1}^{n} \varepsilon_i)^p, 0 < p \leq 1$$

.

**Lemma 2** ([12]). Exist $0 < q < 1$ for $\forall x_1 \geq x_2 \geq 0$, there is

$$x_2(x_1 - x_2)^{q+1} \leq (x_1^{q+2} - x_2^{q+1}) \frac{q+1}{q+2}$$

where $0 < r \leq 1$, and $m = 1, 2, \ldots, n$.

**Lemma 3** ([12]). There is a continuous positive definite function $V(x)$ for constants $\mu_1 > 0$, $\mu_2 > 0$, $0 < \gamma < 1$, $0 < \xi < \infty$. For system $\dot{x} = F(x)$, there exists $\dot{V}(x) \leq -\mu_1 V(x) - \mu_2 V^\gamma(x) + \xi$, then the upper bound on the stabilization time $T_r$ is

$$T_r \leq \max\{t_0 + \frac{1}{\theta_0 \mu_1 (1-\gamma)} \ln(\frac{\theta_0 \mu_1 V^{1-\gamma}(t_0) + \mu_2}{\mu_2}),$$
$$t_0 + \frac{1}{\mu_1 (1-\gamma)} \ln(\frac{\mu_1 V^{1-\gamma}(t_0) + \theta_0 \mu_2}{\theta_0 \mu_2})\}$$

where $0 < \theta_0 < 1$.

## F. Prescribed Performance Control

The formation tracking error is defined as

$$e_i = d(t) - d^*(t) \quad (13)$$

It is clear that the connectivity preservation and collision avoidance constraints are not violated when the tracking error satisfies the following inequality

$$d_{saf}(t) < e_i(t) < d_{com}(t) \quad (14)$$

where $d_{saf}(t) \in \mathbb{R}_+$ denotes the safety constraint distance, $d_{com}(t) \in \mathbb{R}_+$ denotes the compactness constraint distance, and $d_{saf}(t) < d_{com}(t)$.

In control design, to meet the performance criteria for $e_i(t)$, this paper imposes asymmetric performance constraints on the bounds of inequality (13).

$$-\beta_{mi} \rho_i(t) < e_i(t) < \rho_i(t) \quad (15)$$

where $\beta_{mi} \in \mathbb{R}_+^2$, $\rho_i(t)$ denotes a continuous performance function that satisfies $\rho_i(t) > 0$ and is strictly decreasing.

$$\lim_{t \to \infty} \rho_i(t) = \rho_{i,\infty}(t) > 0 \quad (16)$$

The performance function $\rho_i(t)$ is set as an exponential decay function, with its parameters tuned to control the convergence rate and accuracy of the tracking error. The boundary function is defined as

$$\rho_i(t) = (\rho_{i,0}(t) - \rho_{i,\infty}(t))\exp(-k_{mi}t) + \rho_{i,\infty}(t) \qquad (17)$$

where $k_{mi}$, $\rho_{mi,\infty}$, $\rho_{mi,0}$ are positive constants.

The initial state of the vehicle platoon satisfies the following conditions

$$\rho_{i,0}(t) = d_{com}(t) - d^*(t) \qquad (18)$$

$$\beta_{mi} = \frac{d^*(t) - d_{saf}(t)}{d_{com}(t) - d^*(t)} \qquad (19)$$

Converting constrained errors to unconstrained errors to achieve performance metrics, for $\forall t \geq 0$, we have

$$e_i(t) = \rho_i(t)\upsilon(z_i(t)) \qquad (20)$$

where $\upsilon(\cdot)$ is a strictly monotonically increasing smooth function with the expression

$$\upsilon(z_i(t)) = \frac{e^{z_i(t)} - \beta_{mi}e^{-z_i(t)}}{e^{z_i(t)} + e^{-z_i(t)}} \qquad (21)$$

Then

$$\begin{aligned} z_i(t) &= \upsilon^{-1}\left(\frac{e_i(t)}{\rho_i(t)}\right) \\ &= \frac{1}{2}\ln\left(\frac{\rho_i(t) + \beta_{mi}\rho_i(t)}{\rho_i(t) - e_i(t)}\right) \end{aligned} \qquad (22)$$

$$\dot{z}_i(t) = \Lambda_i\left(\dot{e}_i(t) - \frac{\dot{\rho}_i(t)e_i(t)}{\rho_i(t)}\right) \qquad (23)$$

where

$$\Lambda_i = \frac{1}{2}\left(\frac{1}{e_i(t) + \beta_{mi}\rho_i(t)} - \frac{1}{e_i(t) - \rho_i(t)}\right) \qquad (24)$$

and it satisfies $0 < \Lambda_i \leq \bar{\Lambda}_i = \frac{\beta_{mi}+1}{\beta_{mi}\rho_{i,\infty}}$.

Defining the error variable and deriving it with respect to time yields

$$\varepsilon_{di} = z_i(t) - \frac{1}{2}\ln(\beta_{mi}) \qquad (25)$$

From (8),(37) and (39), we have

$$\dot{\varepsilon}_{di} = \Lambda_i\left(v_{i-1}(t) - v_i(t) - h_ia_i(t) - \frac{\rho_i(t)e_i(t)}{\rho_i(t)}\right) \qquad (26)$$

**Remark 1**. If (15) holds, it is easy to verify that (14) holds, the connectivity preservation and collision avoidance problem is transformed into a problem that does not violate the prescribed performance constraints.

## III. CONTROLLER DESIGN

For the vehicle platoon system, this section explains how to develop a finite time control law. The definition of the terminal sliding mode surface $s_i$ as

$$s_i(t) = \dot{z}_i(t) + c_{1i}z_i(t) + c_{2i}|z_i(t)|^{\mu_i}sign(z_i(t)) \qquad (27)$$

where $c_{1i} > 0, c_{2i} > 0$, and $0 < \mu_i < 1$.

Inspired by[6], we introduce a coupled sliding mode surface to manage the relationship between $s_i(t)$ and $s_{i+1}(t)$, thereby ensuring the string stability of the platoon.

$$S_i(t) = \begin{cases} qs_i(t) - s_{i+1}(t), i = 1,2,...,N-1 \\ qs_i(t), i = N \end{cases} \qquad (28)$$

The time derivative of $S_i$ is described as

$$\dot{S}_i(t) = q(\ddot{z}_i + c_{1i}\dot{z}_i + c_{2i}\mu_i|z_i|^{\mu_i-1}) - \dot{s}_{i+1}(t) \qquad (29)$$

The time derivative of $S_n$ is described as

$$\dot{S}_n(t) = q(\ddot{z}_n + c_{1n}\dot{z}_n + c_{2n}\mu_n|z_n|^{\mu_n-1}) \qquad (30)$$

Based on (27) (28) and (29), the derivative of $S_i(t)$ can be written as

$$\begin{aligned} \dot{S}_i &= q\Lambda_i\ddot{e}_i + q[\Lambda_i\frac{(\dot{e}_i\rho_i + e_i\ddot{\rho}_i)\rho_i - e_i\dot{\rho}_i^2}{-\rho_i^2} + \dot{\Lambda}_i(\dot{e}_i - \frac{\dot{\rho}_ie_i}{\rho_i})] \\ &\quad + q(c_{2i}\rho_i|z_i|^{\rho_i-1}\dot{s}_i + c_{1i}\dot{s}_i) - \dot{s}_{i+1} \\ &= -q\Lambda_ih_i\dot{a}_i + A_i \end{aligned} \qquad (31)$$

with $A_i$ is given by:

$$A_i = \begin{cases} q[\Lambda_i(a_{i-1}(t) - a_i(t)) - \Lambda_i\frac{(\dot{e}_i(t))\dot{\rho}_i(t) + e_i(t)\ddot{\rho}_i(t))\rho_i(t) - e_i(t)\dot{\rho}_i^2(t)}{\rho_i^2(t)} \\ + \dot{\Lambda}_i(\dot{e}_i(t) - \frac{\dot{\rho}_i(t)e_i(t)}{\rho_i(t)}) + c_{1i}\dot{s}_i(t) + c_{2i}\mu_i|z_i(t)|^{\mu_i-1}\dot{s}_i(t)] \\ - \dot{s}_{i+1}(t), for i = 1,2,...,N-1 \\ q[\Lambda_i(a_{i-1}(t) - a_i(t)) - \Lambda_i\frac{(\dot{e}_i(t))\dot{\rho}_i(t) + e_i(t)\ddot{\rho}_i(t))\rho_i(t) - e_i(t)\dot{\rho}_i^2(t)}{\rho_i^2(t)} \\ + \dot{\Lambda}_i(\dot{e}_i(t) - \frac{\dot{\rho}_i(t)e_i(t)}{\rho_i(t)}) + c_{1i}\dot{s}_i(t) + c_{2i}\mu_i|z_i(t)|^{\mu_i-1}\dot{s}_i(t)] \\ , for i = N \end{cases} \qquad (32)$$

The adaptation laws are designed as

$$\begin{aligned} \dot{\hat{\varepsilon}}_i &= [\Upsilon_iS_i\tanh(\frac{S_i}{\gamma_i}) - \xi_{\varepsilon_i}\hat{\varepsilon}_i^{\sigma_i+1} - \kappa_{\varepsilon_i}\hat{\varepsilon}_i] \\ \dot{\hat{\theta}}_i &= [\Upsilon_iS_i^2\frac{u_i^2}{2}\eta^T(x_i)\eta(x_i) - \zeta_{\theta_i}\hat{\theta}_i^{\sigma_i+1} - \kappa_{\theta_i}\hat{\theta}_i] \\ \dot{\hat{Q}}_i &= [\Upsilon_iS_iU_isign(U_iS_i) - \zeta_{Q_i}\hat{Q}_i^{\sigma_i+1} - \kappa_{Q_i}\hat{Q}_i] \end{aligned} \qquad (33)$$

Take a note that the nonlinear functions $f_i(v_i,a_i)(i = 1,...,n)$ in (2) could be unknown. In accordance with Lemma 1, the NN method is used in order to approximatively represent these unidentified functions as

$$f_i(v_i,a_i) = \omega_i^T\eta(x_i) + \varsigma_i \qquad (34)$$

Where $\eta(x_i)$ serve as the fuzzy basis function, $\omega_i^T$ represent the optimal parameter vector, and $\varsigma_i$ denote the approximation error, which is bounded above by an unknown positive constant $\varsigma_i^*$. The goal is to converge the state variables toward the sliding mode surface delineated in equation (28). And $0 < \sigma_i < 1$, $\lambda_{ji}(j = 1,2,3)$ are positive.

$$\dot{S}_i = -\lambda_{2i}S_i - \lambda_{3i}sign(S_i) - \lambda_{1i}|S_i|^{\sigma_i}sign(S_i) \qquad (35)$$

Based on (35), the control law for vehicle $i$ is developed as

$$u_i = Q_i U_i, \ Q_i = \frac{m_i \tau_i}{\Xi_{ki}}$$
$$\varepsilon_i \geq d_i^* + \varsigma_i^* + r_i^*, \theta_i = ||d_i||^2 = d_i^T d_i$$
$$U_i = \frac{1}{\Upsilon_i}[\lambda_{1i}|S_i|^{\sigma_i} sign(S_i) + \lambda_{2i}S_i + \lambda_{3i}sign(S_i)$$
$$+ A_i + \Upsilon_i \frac{\iota_i^2}{2}\hat{\theta}_i \eta^T(x_i)\eta(x_i)S_i + \Upsilon_i \hat{\varepsilon} \tanh(\frac{S_i}{\gamma_i})] \tag{36}$$

where $\Upsilon_i = qh\Lambda_i$.

Proof: The proof is divided into two parts.

Step 1. Consider the Lyapunov function

$$V_{1i} = \frac{1}{2}S_i^2 + \frac{1}{2}\tilde{\theta}_i^2 + \frac{1}{2}\tilde{\varepsilon}_i^2 + \frac{1}{2}\tilde{Q}_i^2 \tag{37}$$

where $(\tilde{\bullet}) = (\bullet) - (\hat{\bullet})$.

The time derivative of $V_{1i}$ is described as

$$\dot{V}_{1i} = S_i\dot{S}_i + \tilde{\theta}_i\dot{\tilde{\theta}}_i + \tilde{\varepsilon}_i\dot{\tilde{\varepsilon}}_i + \tilde{Q}_i\dot{\tilde{Q}}_i \tag{38}$$

Based on (35) and (36), we have

$$\dot{S}_i = -q\Lambda_i h_i \dot{a}_i + A_i$$
$$= -[\Upsilon_i \frac{\iota_i^2}{2}\hat{\theta}_i \eta^T(x_i)\eta(x_i)S_i + \Upsilon_i \hat{\varepsilon} \tanh(\frac{S_i}{\gamma_i})$$
$$+ \lambda_{1i}|S_i|^{\sigma_i}sign(S_i) + \lambda_{2i}S_i + \lambda_{3i}sign(S_i)] \tag{39}$$
$$+ \Upsilon_i \omega_i^T \eta(x_i) - \Upsilon_i[d_i(t) + \varsigma_i + \frac{r_i(t)}{m_i\tau_i}] + \Upsilon_i \frac{\Xi_i}{m_i\tau_i}\tilde{Q}_i U_i$$

Further, we can derive that

$$S_i\dot{S}_i \leq -\lambda_{1i}|S_i|^{\sigma_i+1}sign(S_i) - \lambda_{3i}S_i sign(S_i)$$
$$- \Upsilon_i S_i \frac{\iota_i^2}{2}\hat{\theta}_i \eta^T(x)\eta(x)S_i + \Upsilon_i S_i \omega_i^T \eta(x_i)$$
$$- \lambda_{2i}S_i^2 + \Upsilon_i S_i(\varepsilon - \hat{\varepsilon}\tanh(\frac{S_i}{\gamma_i})) + \Upsilon_i S_i \frac{\Xi_{ki}}{m_i\tau_i}\tilde{Q}_i U_i \tag{40}$$

There exists satisfying

$$\Upsilon_i S_i \omega_i^T \eta(x_i) \leq \Upsilon_i S_i \frac{\iota_i^2}{2}\theta_i \eta^T(x)\eta(x)S_i + \Upsilon_i \frac{1}{2\iota_i^2} \tag{41}$$

where $\iota_i > 0$, and

$$S_i\dot{S}_i \leq -\lambda_{1i}|S_i|^{\sigma_i+1}sign(S_i) - \lambda_{3i}S_i sign(S_i)$$
$$- \lambda_{2i}S_i^2 + \Upsilon_i S_i^2 \frac{\iota_i^2}{2}(\theta_i - \hat{\theta}_i)\eta^T(x)\eta(x)$$
$$+ \Upsilon_i \frac{1}{2\iota_i^2} + \Upsilon_i S_i(\varepsilon - \hat{\varepsilon}\tanh(\frac{S_i}{\gamma_i})) + \Upsilon_i S_i \frac{\Xi_{ki}}{m_i\tau_i}\tilde{Q}_i U_i \tag{42}$$

Similarly, we have

$$\tilde{\theta}_i\dot{\tilde{\theta}} \leq -\tilde{\theta}_i\Upsilon_i S_i^2 \frac{\iota_i^2}{2}\eta^T(x_i)\eta(x_i) + \zeta_{\theta_i}\tilde{\theta}_i\hat{\theta}_i^{\sigma_i+1} + \kappa_{\theta_i}\tilde{\theta}_i\hat{\theta}_i \tag{43}$$

$$\tilde{\varepsilon}_i\dot{\tilde{\varepsilon}}_i \leq -\tilde{\varepsilon}_i\Upsilon_i S_i \tanh(\frac{S_i}{\gamma_i}) + \zeta_{\varepsilon_i}\tilde{\varepsilon}_i\hat{\varepsilon}_i^{\sigma_i+1} + \kappa_{\varepsilon_i}\tilde{\varepsilon}_i\hat{\varepsilon}_i \tag{44}$$

and

$$\tilde{Q}_i\dot{\tilde{Q}}_i \leq -\tilde{Q}_i\Upsilon_i S_i U_i sign(U_i S_i) + \zeta_{Q_i}\tilde{Q}_i\hat{Q}_i^{\sigma_i+1} + \kappa_{Q_i}\tilde{Q}_i\hat{Q}_i \tag{45}$$

Substituting (41)-(45) into (38) yields

$$\dot{V}_{1i} = S_i\dot{S}_i + \tilde{\theta}_i\dot{\tilde{\theta}}_i + \tilde{\varepsilon}_i\dot{\tilde{\varepsilon}}_i + \tilde{Q}_i\dot{\tilde{Q}}_i$$
$$\leq -\lambda_{1i}|S_i|^{\sigma_i+1}sign(S_i) - \lambda_{2i}S_i^2 - \lambda_{3i}S_i sign(S_i)$$
$$+ \Upsilon_i S_i^2 \frac{\iota_i^2}{2}(\theta_i - \hat{\theta}_i)\eta^T(x)\eta(x) + \Upsilon_i \frac{1}{2\iota_i^2}$$
$$- \tilde{\varepsilon}_i\Upsilon_i S_i \tanh(\frac{S_i}{\gamma_i}) + \zeta_{\varepsilon_i}\tilde{\varepsilon}_i\hat{\varepsilon}_i^{\sigma_i+1} + \kappa_{\varepsilon_i}\tilde{\varepsilon}_i\hat{\varepsilon}_i \tag{46}$$
$$+ \zeta_{Q_i}\tilde{Q}_i\hat{Q}_i^{\sigma_i+1} + \kappa_{Q_i}\tilde{Q}_i\hat{Q}_i + \Upsilon_i S_i \frac{\Xi_{ki}}{m_i\tau_i}\tilde{Q}_i U_i$$
$$+ \Upsilon_i S_i(\varepsilon_i - \hat{\varepsilon}_i\tanh(\frac{S_i}{\gamma_i})) - \tilde{Q}_i\Upsilon_i S_i U_i sign(U_i S_i)$$
$$- \tilde{\theta}_i\Upsilon_i S_i^2 \frac{\iota_i^2}{2}\eta^T(x_i)\eta(x_i) + \zeta_{\theta_i}\tilde{\theta}_i\hat{\theta}_i^{\sigma_i+1} + \kappa_{\theta_i}\tilde{\theta}_i\hat{\theta}_i$$

The following equation is maintained given that $q$, $h$, $R_i$, and $Q_i$ are all non-negative.

$$\Upsilon_i S_i \frac{\Xi_{ki}}{m_i\tau_i}\tilde{Q}_i U_i - \tilde{Q}_i\Upsilon_i S_i U_i sign(U_i S_i) \leq 0 \tag{47}$$

Then, we have

$$\dot{V}_{1i} \leq -\lambda_{1i}|S_i|^{\sigma_i+1}sign(S_i) - \lambda_{2i}S_i^2 - \lambda_{3i}S_i sign(S_i)$$
$$+ \zeta_{Q_i}\tilde{Q}_i\hat{Q}_i^{\sigma_i+1} + \kappa_{Q_i}\tilde{Q}_i\hat{Q}_i + \zeta_{\varepsilon_i}\tilde{\varepsilon}_i\hat{\varepsilon}_i^{\sigma_i+1} + \kappa_{\varepsilon_i}\tilde{\varepsilon}_i\hat{\varepsilon}_i$$
$$+ \zeta_{\theta_i}\tilde{\theta}_i\hat{\theta}_i^{\sigma_i+1} + \kappa_{\theta_i}\tilde{\theta}_i\hat{\theta}_i + \Upsilon_i \frac{1}{2\iota_i^2} + \Upsilon_i\varepsilon_i(S_i - S_i\tanh(\frac{S_i}{\gamma_i}))$$
$$\tag{48}$$

By using $0 \leq |\chi| - \chi\tanh(\chi/\gamma) \leq \nu\gamma$.

$$\Upsilon_i\varepsilon_i(S_i - S_i tanh(\frac{S_i}{\gamma_i}) \leq 0.2785\Upsilon_i\varepsilon_i\gamma_i \tag{49}$$

where $\nu = 0.2785$ for $\forall \gamma > 0$ and $\chi \in \mathbb{R}$.

Then, we hzve

$$\dot{V}_{1i} \leq -\lambda_{1i}|S_i|^{\sigma_i+1}sign(S_i) - \lambda_{2i}S_i^2 - \lambda_{3i}S_i sign(S_i)$$
$$+ \zeta_{Q_i}\tilde{Q}_i\hat{Q}_i^{\sigma_i+1} + \kappa_{Q_i}\tilde{Q}_i\hat{Q}_i + \zeta_{\varepsilon_i}\tilde{\varepsilon}_i\hat{\varepsilon}_i^{\sigma_i+1} + \kappa_{\varepsilon_i}\tilde{\varepsilon}_i\hat{\varepsilon}_i$$
$$+ \zeta_{\theta_i}\tilde{\theta}_i\hat{\theta}_i^{\sigma_i+1} + \kappa_{\theta_i}\tilde{\theta}_i\hat{\theta}_i + \Upsilon_i(0.2785\varepsilon_i\gamma_i + \frac{1}{2\iota_i^2}) \tag{50}$$

Based on Lemma 2, we have

$$\zeta_{\theta_i}\tilde{\theta}_i\hat{\theta}_i^{\sigma_i+1} + \kappa_{\theta_i}\tilde{\theta}_i\hat{\theta}_i$$
$$\leq \zeta_{\theta_i}\frac{\sigma_i+1}{\sigma_i+2}(\theta_i^{\sigma_i+2} - \tilde{\theta}_i^{\sigma_i+1}) + \frac{\kappa_{\theta_i}}{2}(\theta_i^2 - \tilde{\theta}_i^2) \tag{51}$$

$$\zeta_{\varepsilon_i}\tilde{\varepsilon}_i\hat{\varepsilon}_i^{\sigma_i+1} + \kappa_{\varepsilon_i}\tilde{\varepsilon}_i\hat{\varepsilon}_i$$
$$\leq \zeta_{\varepsilon_i}\frac{\sigma_i+1}{\sigma_i+2}(\varepsilon_i^{\sigma_i+2} - \tilde{\varepsilon}_i^{\sigma_i+1}) + \frac{\kappa_{\varepsilon_i}}{2}(\varepsilon_i^2 - \tilde{\varepsilon}_i^2) \tag{52}$$

$$\zeta_{Q_i}\tilde{Q}_i\hat{Q}_i^{\sigma_i+1} + \kappa_{Q_i}\tilde{Q}_i\hat{Q}_i$$
$$\leq \zeta_{Q_i}\frac{\sigma_i+1}{\sigma_i+2}(Q_i^{\sigma_i+2} - \tilde{Q}_i^{\sigma_i+1}) + \frac{\kappa_{Q_i}}{2}(Q_i^2 - \tilde{Q}_i^2) \tag{53}$$

Substituting (51)-(53) into (50), we have

$$V_{1i} \leq -2^{\sigma_i+1} \min\left\{\lambda_{1i}, \zeta_{\theta_i} \frac{\sigma_i+1}{\sigma_i+2}, \zeta_{\varepsilon_i} \frac{\sigma_i+1}{\sigma_i+2}, \zeta_{Q_i} \frac{\sigma_i+1}{\sigma_i+2}\right\}$$
$$\left\{\left(\tfrac{1}{2}S_i^2\right)^{\bar{\sigma}_i} + \left(\tfrac{1}{2}\tilde{\varepsilon}_i^2\right)^{\bar{\sigma}_i} + \left(\tfrac{1}{2}\tilde{\theta}_i^2\right)^{\bar{\sigma}_i} + \left(\tfrac{1}{2}\tilde{Q}_i^2\right)^{\bar{\sigma}_i}\right\}$$
$$- \min\left\{2\lambda_{2i}, \kappa_{\theta_i}, \kappa_{\varepsilon_i}, \kappa_{Q_i}\right\}\left\{\tfrac{1}{2}S_i^2 + \tfrac{1}{2}\tilde{\varepsilon}_i^2 + \tfrac{1}{2}\tilde{\theta}_i^2 + \tfrac{1}{2}\tilde{Q}_i^2\right\}$$
$$+ \zeta_{\varepsilon_i} \frac{\sigma_i+1}{\sigma_i+2}\varepsilon_i^{\sigma_i+2} + \frac{\kappa_{\varepsilon_i}}{2}\varepsilon_i^2 + \zeta_{\theta_i} \frac{\sigma_i+1}{\sigma_i+2}\theta_i^{\sigma_i+2} + \frac{\kappa_{\theta_i}}{2}\theta_i^2$$
$$+ \zeta_{Q_i} \frac{\sigma_i+1}{\sigma_i+2}Q_i^{\sigma_i+2} + \frac{\kappa_{Q_i}}{2}Q_i^2 + \Upsilon_i(0.2785\varepsilon_i\gamma_i + \tfrac{1}{2t_i^2})$$
$$\leq -\Upsilon_{1i}V_{1i}^{\frac{\sigma_i+1}{2}} - \Upsilon_{2i}V_{1i} + \Delta_i \tag{54}$$

where
$$\Upsilon_{1i} = 2^{\sigma_i+1} \min\left\{\lambda_{1i}, \zeta_{\theta_i} \frac{\sigma_i+1}{\sigma_i+2}, \zeta_{\varepsilon_i} \frac{\sigma_i+1}{\sigma_i+2}, \zeta_{Q_i} \frac{\sigma_i+1}{\sigma_i+2}\right\}$$
$$\Upsilon_{2i} = \min\left\{2\lambda_{2i}, \kappa_{\theta_i}, \kappa_{\varepsilon_i}, \kappa_{Q_i}\right\}$$
$$\Delta_i = \zeta_{Q_i} \frac{\sigma_i+1}{\sigma_i+2}Q_i^{\sigma_i+2} + \frac{\kappa_{Q_i}}{2}Q_i^2 + \zeta_{\theta_i} \frac{\sigma_i+1}{\sigma_i+2}\theta_i^{\sigma_i+1} + \frac{\kappa_{\theta_i}}{2}\theta_i^2$$
$$+ \zeta_{\varepsilon_i} \frac{\sigma_i+1}{\sigma_i+2}\varepsilon_i^{\sigma_i+2} + \frac{\kappa_{\varepsilon_i}}{2}\varepsilon_i^2 + \Upsilon_i(0.2785\varepsilon_i\gamma_i + \tfrac{1}{2t_i^2})$$

It is evident from [12] that $V_{1i}$ is nearly finite time stable. This indicates that $V_{1i}$ can reach the area $\Omega_i$ in limited time if there is a constant $\bar{\theta}_i$.

$$\Omega_i = \left\{\lim_{t\to T_{1i}} V_{1i} \leq \min\left\{\left(\frac{\Delta_i}{(1-\bar{\theta}_i)\Upsilon_{1i}}\right), \left(\frac{\Delta_i}{(1-\bar{\theta}_i)\Upsilon_{2i}}\right)^{\frac{1}{\bar{\sigma}_i}}\right\}\right\} \tag{55}$$

and $T_{1i}$ satisfies

$$T_{1i} \leq T_{1i\max} := \left\{\frac{1}{\bar{\theta}_i\Upsilon_{1i}(1-\bar{\sigma}_i)} \ln \frac{\bar{\theta}_i\Upsilon_{1i}V^{1-\bar{\sigma}_i}(0)+\Upsilon_{2i}}{\Upsilon_{2i}}, \frac{1}{\Upsilon_{1i}(1-\bar{\sigma}_i)} \ln \frac{\Upsilon_{1i}V^{1-\bar{\sigma}_i}(0)+\bar{\theta}_i\Upsilon_{2i}}{\bar{\theta}_i\Upsilon_{2i}}\right\} \tag{56}$$

The errors $S_i$, $\tilde{\varepsilon}_i$, and $\tilde{\theta}_i$ has been proven through their bounds. Given the inequality $\tfrac{1}{2}S_i^2 \leq V_{S_i} \leq V_{1i}$, it has been deduced that the errors $S_i$ will reach a designated region, characterized by $|S_i| \leq \min\{\sqrt{\frac{\Delta_i}{(1-\bar{\theta}_i)\Upsilon_{1i}}}, \sqrt{(\frac{\Delta_i}{(1-\bar{\theta}_i)\Upsilon_{2i}})^{\frac{1}{\bar{\sigma}_i}}}\}$, within a finite time frame. This ensures that these errors will remain confined within this specified area within a defined time limit, thereby confirming their practical finite-time stability.

Step 2: *Stability analysis during the sliding phase:* By selecting suitable design parameters, $\mu_i$ can be made to converge to an arbitrarily small neighborhood. When $S_i \approx 0$ for $t \geq T_{1i}$, it follows that $s_i = 0$. Consequently, the sliding mode surface $s_i$ given in equation (27) can be reformulated as

$$\dot{s}_i = -c_{1i}z_i - c_{2i}|z_i|^{\mu_i}sign(z_i) \tag{57}$$

Consider the Lyapunov function

$$V_{2i} = \frac{1}{2}z_i^2 \tag{58}$$

Then

$$\dot{V}_{2i} = -2^{\frac{\rho_i+1}{2}}c_{2i}V_{2i}^{\frac{\rho_i+1}{2}} - 2c_{1i}V_{2i} \tag{59}$$

Therefore, leveraging the finite-time theorem from [7], we establish the global finite-time stability of $V_{2i}$, and the spacing error converges to zero within a finite time $T_{2i}$.

$$T_{2i} \leq T_{2i\max} := \frac{1}{c_{1i}(1-\rho_i)}\ln\left(\frac{2c_{1i}V_{2i}^{\frac{\rho_i+1}{2}}(0)+2^{\frac{\rho_i+1}{2}}c_{2i}}{2^{\frac{\rho_i+1}{2}}c_{2i}}\right) \tag{60}$$

The analysis presented shows that the tracking error converges to a small neighborhood around zero for $T_i \geq T_{1i} + T_{2i}$.

Theorem 2: The string stability (12) of the vehicular platoon system, as stated in Theorem 1, is ensured by the control law developed in (36) for all $0 < q \leq 1$.

Proof: The proof of string stability follows a similar approach to that in [12]. Given the definition in (28) where $S_i(t) = qs_i(t) - s_{i+1}(t)$, and with $\mu_i$ converging to a small region near the origin by selecting suitable design parameters, we have

$$qs_i(t) - s_{i+1}(t) \approx 0 \tag{61}$$

It is obvious that $\frac{s_{i+1}(t)}{s_i(t)} \approx q$. Since $0 < q \leq 1$, then $0 < \frac{s_{i+1}(t)}{s_i(t)} \leq 1$.

Employing the sign preservation property of the Peer Mapping theorem, we note that the error $z_i(t)$ and the sliding surface $s_i(t)$ maintain the same sign, which leads to $s_i(t)z_i(t) \geq 0$. This in turn suggests that $z_i(t)z_{i+1}(t) \geq 0$, considering that $s_{i+1}(t)s_i(t) \geq 0$. Given that $0 < s_{i+1}(t)/s_i(t) \approx q \leq 1$, and in accordance with equation (27), $0 < z_{i+1}(t)/z_i(t) \leq 1$. The proof is by contradiction. Assuming against our assertion that $z_{i+1}(t)/z_i(t) > 1$.

1) Let $0 < z_i(t) < z_{i+1}(t)$. The sliding mode surface (27) expressed as

$$s_i(t) = \dot{z}_i(t) + c_{1i}z_i + c_{2i}|z_i|^{\mu_i} \tag{62}$$

From $0 < z_i(t) < z_{i+1}(t)$ and $z_{i+1}(t)/z_i(t) > 1$, it is obvoious that $z_i(t)e^{-st} < z_{i+1}(t)e^{-st}$. Then $0 < \int_0^t z_i(t)e^{-st}dt < \int_0^t z_{i+1}(t)e^{-st}dt$. The $L_i(s) = \int_0^t z_i(t)e^{-st}dt$ is the Laplace transform of $z_i(t)$, then $0 < L_i(s) < L_{i+1}(s)$, we can get $c_{2i}L_{i+1}^{\mu_i} > c_{2i}L_i^{\mu_i}$. Then, we have

$$sL_{i+1}(s) + c_{1i}L_{i+1} + c_{2i}L_{i+1}^{\mu_i} > sL_i(s) + c_{1i}L_i + c_{2i}L_i^{\mu_i} > 0 \tag{63}$$

Further, we have

$$\dot{z}_{i+1}(t) + c_{1i}z_{i+1} + c_{2i}z_{i+1}^{\mu_i} > \dot{z}_i(t) + c_{1i}z_i + c_{2i}z_i^{\mu_i} > 0 \tag{64}$$

Therefore $s_{i+1}(t)/s_i(t) > 1$, it is contrary to the reality $0 < s_{i+1}(t)/s_i(t) \approx q \leq 1$. Since then, we have $0 < z_{i+1}(t)/z_i(t) \approx q \leq 1$.

2) The sliding mode surface (27) may be recast as follows for the case when $z_{i+1}(t) < z_i(t) < 0$.

$$s_i(t) = \dot{z}_i + c_{1i}z_i - c_{2i}|z_i|^{\mu_i} \tag{65}$$

From $z_{i+1}(t) < z_i(t) < 0$ and $|z_{i+1}(t)|/|z_i(t)| > 1$, it is obvious that $z_{i+1}(t)e^{-st} < z_i(t)e^{-st}$. Then, we have $\int_0^t z_{i+1}(t)e^{-st}dt < \int_0^t z_i(t)e^{-st}dt < 0$. The Laplace transform of

$z_i(t)$ is $L_i(s) = \int_0^t z_i(t)e^{-st}dt$, then $L_{i+1}(s) < L_i(s) < 0$, we can get $c_{2i}|L_{i+1}(s)|^{\mu_i} < c_{2i}|L_i(s)|^{\mu_i}$.

Then, we have

$$sL_{i+1}(s) - c_{2i}|L_{i+1}|^{\mu_i} + c_{1i}L_{i+1} \\ < sL_i(s) - c_{2i}|L_i|^{\mu_i} + c_{1i}L_i < 0 \tag{66}$$

Similarly, we have

$$\dot{z}_{i+1}(t) - c_{2i}|z_{i+1}|^{\mu_i} + c_{1i}z_{i+1} \\ < \dot{z}_i(t) - c_{2i}|z_i|^{\mu_i} + c_{1i}z_i < 0 \tag{67}$$

Consequently, the inequality $s_{i+1}(t)/s_i(t) > 1$ is in disagreement with the known constraint $0 < s_{i+1}(t)/s_i(t) \approx q \leq 1$. This implies that the correct relationship must be $0 < z_{i+1}(t)/z_i(t) \approx q \leq 1$.

According to above analysis, we have

- Since $0 < z_{i+1}(t) \leq z_i(t)$, $0 < \int_0^t z_{i+1}(t)e^{-st}dt \leq \int_0^t z_i(t)e^{-st}dt$, and $0 < L_{i+1}(s) \leq L_i(s)$, it follows naturally. Consequently, $||G_i(s)|| \leq 1$ is satisfied by the transfer function of the error $G_i(s) = L_{i+1}(s)/L_i(s)$.
- $\int_0^t z_i(t)e^{-st}dt \leq \int_0^t z_{i+1}(t)e^{-st}dt < 0$ is evident when $z_i(t) \leq z_{i+1}(t) < 0$, which implies $L_i(s) \leq L_{i+1}(s) < 0$. Hence, $||G_i(s)|| \leq 1$ is satisfied by the transfer function of the error, which is $G_i(s) =: L_{i+1}(s)/L_i(s)$.

From the preceding analysis, we can conclude that $G_i(s) = \frac{L_{i+1}(s)}{L_i(s)} = q$. Therefore, when $0 < q \leq 1$, string stability is ensured.

## IV. SIMULATION STUDIES

To assess the efficacy of the control strategy developed, simulations were conducted in MATLAB for a platoon consisting of $N = 3$ vehicles.

The external disturbances were modeled as $d_i(t) = 0.2\sin(t)$. The fault tolerant model was constructed with $u_{ai}(t) = 0.75 + 0.8\sin(t)$. The actuator fault parameters were selected as $r_{ki} = 0.01\sin(t)$. The prescribed performance function was defined as $\rho_i = (14 - 0.8)e^{-0.95t} + 0.8$. The parameters for the simulation were set as

TABLE I:
PARAMETERS

| $\Delta_i(m)$ | $h_i(s)$ | $A_i(m^2)$ | $\rho_a$ | $g(m/s^2)$ | $C_{ai}$ |
|---|---|---|---|---|---|
| 8 | 12.1 | 2 | 0.2 | 9.8 | 0.2 |

TABLE II:
PARAMETERS

| $c_{1i}$ | $c_{2i}$ | $\mu_i$ | $\sigma_i$ | $\rho_i$ | $\xi_{\theta_i}$ | $\xi_{\varepsilon_i}$ | $\xi_{Q_i}$ |
|---|---|---|---|---|---|---|---|
| 1 | 1 | 0.45 | 0.4 | 0.8 | 0.001 | 0.001 | 0.001 |
| $\lambda_{1i}$ | $\lambda_{2i}$ | $\lambda_{3i}$ | $q$ | $\kappa_{\varepsilon_i}$ | $\kappa_{\theta_i}$ | $\kappa_{\varepsilon_i}$ | $\kappa_{Q_i}$ |
| 40 | 40 | 40 | 0.9 | 0.4 | 0.001 | 0.001 | 0.001 |

Leader vehicle acceleration is set as follows

$$\alpha_0 = \begin{cases} 0\,m/s^2 & t < 2 \\ -1\,m/s^2 & 2 \leq t \leq 3 \\ 0.3\,m/s^2 & 20 \leq t < 35 \\ 0\,m/s^2 & t \geq 35 \end{cases} \tag{68}$$

TABLE III:
PARAMETERS

| i | 1 | 2 | 3 |
|---|---|---|---|
| $m_i(*10^3\text{kg})$ | 2.1 | 1.75 | 1.82 |
| $\tau_i(s)$ | 0.5 | 0.4 | 0.42 |
| $L_i(m)$ | 4 | 4.5 | 4..2 |

TABLE IV:
INITIAL STATES FOR VEHICLES

| i | 0 | 1 | 2 | 3 |
|---|---|---|---|---|
| $p_i(0)(m)$ | 160.4 | 145 | 128.48 | 112.4 |
| $v_i(0)(m/s)$ | 1 | 1 | 1.5 | 2 |
| $a_i(0)(m/s^2)$ | 0 | 1.5 | 2 | 1.8 |

The simulation outcomes are depicted in Figures 1-5. The tracking errors inside the vehicle platoon converge to the preset zone, as shown in Figure 1, indicating that string stability has been achieved. Figure 2 demonstrates that the following vehicle is able to track the preceding one without any collision occurring. Figures 3 and 4 reveal that the velocities and accelerations of all followers successfully track those of the leader under the proposed control method. Figure 5 presents the control inputs, which stabilize around a region near zero. A preliminary analysis indicates that the formulated formation control signals are capable of compensating for intermittent actuator failures in system (2) and maintain the stability of the controlled system (2).

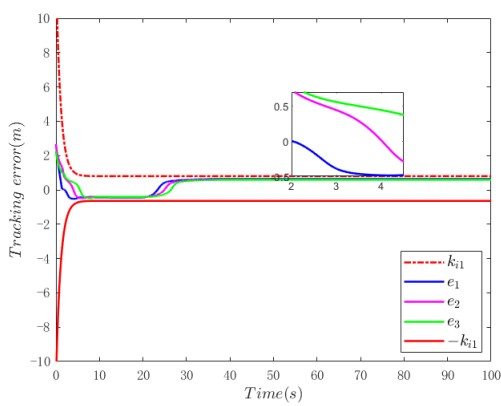

Fig. 1: The tracking error with PPC.

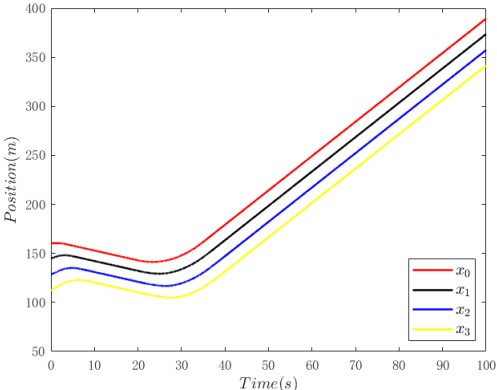

Fig. 2: The positions of followers and leader.

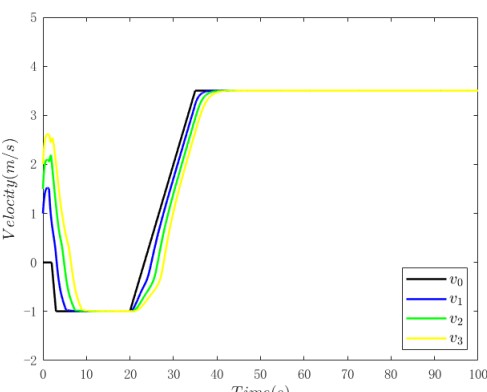

Fig. 3: The velocities of followers and leader.

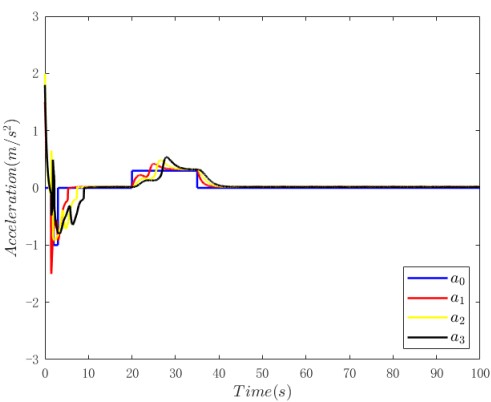

Fig. 4: The accelerations of followers and leader.

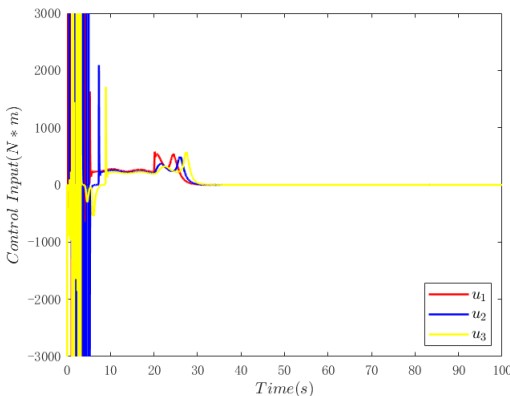

Fig. 5: The control input

## V. CONCLUSIONS

The proposed control scheme enforces performance bounds on the tracking error of the vehicle platoon and ensures the preservation of connection and the avoidance of accidents between neighboring vehicles. The control strategy proposed ensures several key objectives: maintaining connectivity and preventing collisions between adjacent vehicles, enforcing performance constraints on the tracking error within the vehicle platoon, ensuring individual vehicles stability, and achieving string stability over a finite time period. Additionally, the strategy mitigates the impact of actuator failures. Simulation results confirm the effectiveness of this approach. It's essential to highlight that the convergence time of the algorithm is influenced by the initial state of the system, which might limit its applicability in certain scenarios. Therefore, future research efforts should focus on developing fixed-time control solutions tailored specifically for vehicle platoons. These solutions would offer predictable convergence times irrespective of initial conditions, enhancing the robustness and reliability of the control strategy in practical applications.

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
