# OpenReview forum: "Adaptive Fuzzy Fault-Tolerant Formation Control for Third-Order Heterogeneous Vehicle Platoon System with Intermittent Actuator Faults"
_IEEE.org/ICIST/2024/Conference — IEEE ICIST 2024 Conference Submission_

### Official Review · Reviewer_VAYS · 2024-08-22
**This paper is well written and acceptable.**

**Rating:** 9
**Confidence:** 5

**Review:**

1.The engineering background of the proposed problem should be stated more clearly to readers. The literature review is insufficient. Some recently published papers should be included in the references list.
2.In simulation section, more analysis and descriptions should be given to show the effectiveness of the developed method.
3.To make it easier for readers to understand the novelties of this paper, it would be better for the authors to add the control frame diagram 4.The robustness of the proposed control method should be described.

---

### Official Review · Reviewer_vXCn · 2024-08-29
**This paper can be accepted.**

**Rating:** 9
**Confidence:** 4

**Review:**

This paper studies the problem of prescriptive performance tolerant control of a heterogeneous vehicle platoon with intermittent actuator failures. Elegant fuzzy sliding mode control approach has been proposed. simulation results are given to show the effectiveness of the method. The topic is interesting and the contributions are solid. Detailed comment: There are 3 lemmas in Section II. However, only lemma 1 and lemma 2 are used. So, please explicitly show how lemma 3 is used.

---

### Official Review · Reviewer_b3QU · 2024-08-30
**comment**

**Rating:** 7
**Confidence:** 5

**Review:**

This paper investigates the problem of prescriptive performance tolerant control of a heterogeneous vehicle platoon with intermittent actuator failures. In the reviewer’s opinion, there are some major comments in the paper which should be addressed by the authors:
1.The layout spacing is relatively large, and it is recommended to make modifications. Additionally, it is suggested to supplement the innovation points.
2.The curves of each variable in the simulation diagram are only differentiated by color, without distinguishing by line type, which can easily cause confusion.
3.The current simulation results provide only feasibility verification without sufficient comparisons with the state-of-art literature.

---

### Decision · Program_Chairs · 2024-09-06

Accept (Oral)